# Evaluation of Prenatal Comfort, Sleep, and Quality of Life in Pregnant Women with Cholestasis: A Cross-Sectional Study

**DOI:** 10.3390/healthcare12141399

**Published:** 2024-07-13

**Authors:** Yeşim Yeşil, Ülkin Gündüz, Ayşegül Dönmez, Semir Paşa

**Affiliations:** 1Department of Midwifery, Faculty of Health Sciences, Mardin Artuklu University, 47100 Mardin, Turkey; ulkingunduz@artuklu.edu.tr; 2Department of Midwifery, Faculty of Health Sciences, İzmir Tınaztepe University, 35400 İzmir, Turkey; aysegul.donmez@tinaztepe.edu.tr; 3Medikal Park Çanakkale Hospital, 17020 Çanakkale, Turkey; semirp@hotmail.com

**Keywords:** comfort, intrahepatic cholestasis of pregnancy, prenatal, quality of life, sleep quality

## Abstract

Background: Associated with adverse pregnancy outcomes, intrahepatic cholestasis of pregnancy is the most prevalent liver disease that women typically experience during pregnancy. This study aimed to evaluate prenatal comfort, sleep, and quality of life in pregnant women with cholestasis. Methods: This cross-sectional study was implemented between November 2022 and June 2023 at Mardin Training and Research Hospital with 150 pregnant women who received a diagnosis of pregnancy-induced intrahepatic cholestasis and agreed to participate. The following tools were utilized to collect data: A personal information form exploring socio-demographic and obstetric characteristics of participants, the Prenatal Comfort Scale (PCS), the Pittsburgh Sleep Quality Index (PSQI), and the World Health Organization Quality of Life-Brief Form (WHOQOL-BREF). Results: The mean age of participants was 27.79 ± 6.33 years. The mean PCS and PSQI scores were 61.20 ± 5.84 and 9.52 ± 3.02, respectively. The mean scores of “physical health, psychological health, social relationships, and environmental health” sub-dimensions in WHOQOL-BREF were 10.63 ± 2.18, 10.48 ± 2.10, 11.31 ± 3.28, and 11.27 ± 2.10, respectively. A significant difference was found for PSQI regarding hospitalization status and change in sleep quality variables (*p* = 0.025 and *p* = 0.035, respectively). Conclusions: Cholestasis of pregnancy creates problems such as pruritus, body image changes, hospitalization, and poor sleep quality in women. This study showed that pregnant women with cholestasis had low levels of sleep quality and quality of life, implying that cholestasis affects their sleep quality, prenatal comfort levels, and quality of life in general. In addition, it is seen that women with this problem do not want to fall pregnant again.

## 1. Introduction

Intrahepatic cholestasis of pregnancy (ICP) is the most common pregnancy-specific liver disease that occurs during pregnancy. It is characterized by generalized pruritus all over the body, more intense on the palms and soles of the feet, which usually starts at the end of the second trimester and in the third trimester [1]. Itching spreads to the whole body over time and serum bile acids and liver enzymes may be increased. Pruritus increases as the gestational week progresses and usually decreases or disappears completely within 48 h after delivery [2].

Although its etiology is not known with certainty, it is thought to develop multifactorially, with effects of sensitivity to environmental, endocrine, and genetic factors [3]. The most sensitive marker in the diagnosis and follow-up of ICP is elevated serum bile acids. Bile acids include primary bile acids, cholic acid, and chenodeoxycholic acid. Bile acid levels can be up to 10 times higher than normal in gestational cholestasis, and levels above 40 mmol/L increase the risk of fetal complications. A blood bile acid level above 410 mmol/L in the fasting state is diagnostic [4]. ICP is observed more frequently in advanced-age pregnancies, multiple pregnancies, pregnant women with a history of ICP in their own or family history, and in winter season. In women with a history of ICP, the rate of ICP in the next pregnancy varies between 80% and 90%. The incidence of ICP also varies according to ethnicity. In studies, it has been reported that the prevalence of ICP is 1.4% among Asian individuals, 5.6% in America, 1.2% in European countries, and 16% in Chile. In Turkey, the incidence rate varies between 0.45% and 1%, although it varies between studies [5,6].

Treatment of ICP aims to relieve maternal pruritus and reduce the risk of fetal morbidity and mortality. The aim of treatment is to decrease bile acid synthesis from hepatocytes, to prevent reabsorption of bile acid from the intestine and kidney and its uptake into hepatic cells, to decrease mitochondrial damage caused by bile acid, and to prevent neutrophil and immune cells from attacking damaged hepatocytes [7]. Drugs used in treatment for this purpose can be used singly or in combination. The most commonly used agent in treatment is ursodeoxylic acid. Drugs such as cholestyramine, rifampicin, S-adenosyl-methionine, sterizine, and vitamin K can be used for therapeutic purposes, although there is no complete evidence for their efficacy [8].

In the literature, it is reported that maternal prognosis of gestational cholestasis is good, but care should be taken in terms of fetal morbidity and mortality. When we look at the maternal effects of ICP, superficial lesions on the skin and mucosa due to pruritus are observed. In addition, pruritus negatively affects the quality of life and social life of the mother by causing insomnia, fatigue, anxiety, stress, nausea, maternal weight loss, and deterioration in body image [9]. Increased levels of bile acids in maternal serum stimulate oxytocin receptors in uterine myometrial tissue and cause preterm labor. The mother’s fear of losing her baby and excessive stress due to itching lead to increased contractions [10]. Close monitoring of uterine contractions, monitoring of fetal health, and frequent evaluation of maternal liver enzymes increase the frequency of pregnancy follow-up. Increased contractions, decreased fetal movements or bradycardia, and increased levels of bile acids and liver enzymes may lead to hospitalization. The change in the social and physical environment, separation from the family, uncertainty and fear of losing the baby that occur with the hospitalization of the mother negatively affect the mother’s quality of life [10,11].

In a systematic review, it was reported that physical symptoms during pregnancy were closely related to sleep quality, psychological problems, and quality of life [11]. According to another study conducted on patients diagnosed with cholestasis, 74% (120/162) of the participants reported that itching affected their sleep and 22% (19/88) reported that itching was resistant to treatment [12]. In the literature, studies on cholestasis of pregnancy are generally quantitative laboratory studies on pregnant and fetal health [13]. This study was planned because there are limited data on prenatal comfort, sleep, and quality of life in pregnant women diagnosed with intrahepatic cholestasis in pregnancy.

## 2. Materials and Methods

### 2.1. Study Design

This cross-sectional study was conducted between 1 November 2022 and 30 June 2023.

### 2.2. Population and Sample

The study population comprised pregnant women with cholestasis who applied to the gynecology outpatient clinic of a training and research hospital. Since there were no similar studies in the literature, the sample calculation was based on the effect size. When the effect size for the difference between patients with and without change in sleep quality for PSQI was taken as 0.5, a sample of 67 per group and a minimum sample of 134 in total was found sufficient for the study when the sample calculation was made using the Mann–Whitney U test with a power of 0.80 at the 0.05 significance level.

In this context, 150 pregnant women who were diagnosed with cholestasis were reached within the date range when the research data were collected.


*Inclusion criteria were as follows:*
Diagnosed with cholestasis,20th week of gestation and above,Can be communicated with,Agreed to participate in the study.



*Exclusion criteria were as follows:*
Healthy pregnant women,Unable to communicate,Pregnant women who refused to participate in the study.


### 2.3. Ethical Consideration

Ethical approval was obtained from Mardin Artuklu University Non-Interventional Clinical Research Ethics Committee (2022/13-28). In addition, written informed consent was obtained from each pregnant woman before participating in the study.

### 2.4. Data Collection

Created by the researchers in line with the relevant literature, a ‘Participant Information Form’ with 14 questions about participants’ socio-demographic and obstetric characteristics was used in addition to ‘Prenatal Comfort Scale’, ‘Pittsburgh Sleep Quality Index’, and ‘World Health Organization Quality of Life Scale’ to collect data.

*Participant Information Form:* The questionnaire consisted of questions on socio-demographic characteristics (e.g., age and education, employment status, family type), obstetric characteristics (e.g., number of pregnancies, previous mode of delivery, thinking about repeat pregnancy) and cholestasis-related questions (e.g., hospitalization, body image change, poor sleep quality, and impact on social life).

*Prenatal Comfort Scale:* Developed by Takeishi et al. (2011) in Japan [11] and confirmed by Kaya Şenol et al. (2020) in terms of reliability and validity in Turkish, the scale aims to present prenatal comfort. Kaya Şenol et al. (2020) found the internal consistency coefficient of the scale (Cronbach’s Alpha) to be 0.86 [12]. It involves 15 items along with the following sub-dimensions: “The effect of improving relationships with the spouse on the paternal role—Spouse”, “Interacting with the movements of the fetus—Fetus”, “Social support received from the people around—Social Environment”, “Acceptance of the maternal role and attachment to the baby—Motherhood”, and “Recognizing changes in oneself during pregnancy—Pregnancy”. It is a 6-point Likert scale where each item is scored between 0–5 as follows: 0 = Strongly disagree, 1 = Disagree, 2 = Unsure, 3 = Somewhat agree, 4 = Agree, and 5 = Strongly agree. The maximum overall score is 75 with no reverse-scored items and no cut-off point, which means the higher the score, the greater the comfort level.

*Pittsburgh Sleep Quality Index (PSQI):* Developed by Buysse et al. (1989) [13] and confirmed by Ağargün et al. (1996) in terms of reliability and validity in Turkish, the scale intends to assess sleep quality and sleep disturbance within the last month. Ağargün et al. (1996) found the internal consistency coefficient of the scale (Cronbach’s Alpha) to be 0.8 [14]. It includes 24 questions, of which 19 are answered by the respondent and 5 by his/her spouse or roommate. Only the questions answered by the respondent are included in the score calculation. Seven components are obtained out of 19 questions assessed, where each is scored between 0–3 points. Thus, the maximum total score is 21. The higher the score, the poorer the sleep quality. An overall score lower than or equal to 5 implies “good sleep”, whereas the opposite means “poor sleep” [13,14].

*World Health Organization Quality of Life-Brief Form (WHOQOL-BREF):* Developed by the WHO in 1998 and converted to Turkish by Eser et al. (1999) [15] in terms of validity and reliability, this scale (namely, WHOQOL-BREF-TR) was employed to determine the quality of life of the participants. WHOQOL-BREF-TR has 27 items in total including an additional “national” question, since Eser et al. (1999) considered this question to be significantly related to the general health and quality of life of Turkish population. It involves four domains: Physical Health, Psychological Health, Social Relationships, and Environmental Health. The scale does not have a total score, but each domain is assessed on its own. Following preparation of the dataset, scores of the domains were sent to the WHOQOL center in Turkey for calculation. The scores were assessed between 4 and 20 where higher scores implied greater levels of well-being. Eser et al. (1999) found the internal consistency coefficients (Cronbach’s Alpha) to be 0.83 for physical health, 0.66 for psychological health, 0.53 for social relations, and 0.73 for environmental health [15]. In this study, those coefficients were 0.52 in the physical domain, 0.50 in the psychological domain, 0.67 in the social domain, and 0.72 in the environmental domain.

### 2.5. Statistical Analysis

Statistical Package for the Social Sciences (SPSS 11.5 Chicago, IL, USA) was used for data analysis. Mean ± standard deviation and median (minimum–maximum) values were used as descriptors for quantitative variables. Since normal distribution assumptions were not met, the Mann–Whitney U test was used to determine whether there was a difference between categories with two categories in terms of quantitative variables. The Kruskal–Wallis H test was used to examine whether there was a difference between categories with more than two categories in terms of quantitative variables, since normal distribution assumptions were not met. The Spearman correlation coefficient was used to examine the relationship between two quantitative variables, since normal distribution assumptions were unmet. The statistical significance level was taken as *p* < 0.05.

## 3. Results

Descriptive data of the participants are presented in Table 1. Their mean age was 27.79 ± 6.33 years. While 29.3% of them were literate, 44.7% had graduated from primary school and 26.0% from high school or above. Among the patients, 20.0% were actively employed and 59.3% had social security. Most of them (80.7%) had a nuclear family. A majority of them had multiparous pregnancies (81.3%) and a previous vaginal delivery (84.7%).

The mean numbers of gravida, para, abortion, and number of living children were 3.71 ± 2.14, 2.39 ± 1.87, 0.33 ± 0.70, and 2.33 ± 1.87, respectively. The mean weight gain after pregnancy was 13.81 ± 5.34. Additionally, 78% of the pregnant women did not plan to become pregnant again. During pregnancy, 18.7% reported hospitalization, 96% reported body image changes, 98% reported poor sleep quality, and all reported that their social life was affected (Table 1).

For the Prenatal Comfort Scale, the mean scores of sub-dimensions Husband, Fetus, People, Mother, and Myself were 15.94 ± 2.30, 8.49 ± 1.32, 12.12 ± 1.66, 12.35 ± 1.59, and 12.30 ± 1.52, respectively, while the mean total score was 61.20 ± 5.84.

The mean PSQI total score was 9.52 ± 3.02. The mean scores of the Physical Health, Psychological Health, Social Relationships, and Environmental Health domains of the Quality-of-Life Scale were 10.63 ± 2.18, 10.48 ± 2.10, 11.31 ± 3.28, and 11.27 ± 2.10, respectively (Table 2).

For the Prenatal Comfort Scale, there was a negative, low magnitude, and statistically significant relationship between scores of Mother and Myself scales and age (as age increased, Mother and Myself scale scores decreased), and between Myself scale score and weight gained during pregnancy (as the weight gained during pregnancy increased, Myself scale score decreased). Similarly, there was a negative, low magnitude, and statistically significant relationship between PSQI total score and weight gained during pregnancy (as the weight gained during pregnancy increased, the PSQI total score decreased).

On the other hand, a positive, low magnitude, and statistically significant relationship was found between the Environmental Health scale score and age (as age increased, Environmental Health scale score increased) (Table 3).

For the Pittsburgh Sleep Quality Index (PSQI), there was a significant difference in terms of hospitalization status and change in sleep quality (*p* = 0.025 and *p* = 0.035, respectively). The mean PSQI score was 8.36 ± 2.93 in hospitalized pregnant women and 9.79 ± 2.99 in non-hospitalized pregnant women.

Pregnant women with no change in sleep quality had significantly higher PSQI score on average than those with a change in sleep quality (Table 4).

For WHOQOL-BREF-TR, the following variables had a significant difference: previous delivery type, contemplating pregnancy again, hospitalization status, and body image change status in Physical Health sub-dimension (*p* = 0.043, *p* = 0.010, *p* = 0.036, and *p* = 0.028), health assurance status and status of considering pregnancy again in Psychological Health sub-dimension (*p* = 0.007 and *p* = 0.031), status of considering pregnancy again and hospitalization status in Social Relationships sub-dimension (*p* = 0.002 and *p* = 0.019), and hospitalization status in Environmental Health sub-dimension (*p* = 0.042) (Table 5).

## 4. Discussion

One of the high-risk diseases during pregnancy, cholestasis of pregnancy is the most prevalent liver disease that pregnant women face [16]. Prenatal comfort, sleep quality, and quality of life of pregnant women with cholestasis of pregnancy were examined in this study. We have not encountered a study in the literature analyzing quality of life, sleep quality, and prenatal comfort of pregnant women with gestational cholestasis. Hence, our findings were discussed with those of studies on risky pregnancies.

Complaints and different psycho-social factors experienced during pregnancy result in stress, thereby adversely affecting pregnant women’s comfort level. Our research found a significant negative correlation between Mother and Myself sub-dimensions of the Prenatal Comfort scale and age. As age increases, Mother and Myself subscale scores decrease. The findings of Aydın Özkan et al. (2020) support our study [17]. This likely results from the fact that pregnant women in the younger age group are better able to tolerate physical discomfort and other pregnancy-related problems.

There was a significant negative correlation between the Myself scale score and the weight gained during pregnancy. As the weight gained during pregnancy increases, the Myself scale score decreases. Kaplan and Demircan Sezer (2023) found a significant increase in obstetric complication rates in pregnant women with a weight gain of 13 kg or more during pregnancy [18]. Although pregnancy is a physiological process, changes in women’s bodies, such as weight gain, linea nigra, and striae, negatively affect their self-image [5].

It has been reported that excessive weight gain during pregnancy raises the risk of diverse pregnancy complications as well as chronic health issues in the long run and it obviously affects women’s lives in every aspect, as shown by our research and other studies.

The most common symptom associated with gestational cholestasis is pruritus. It typically emerges on palms and soles of the feet but can cover the whole body. It worsens at night and often affects the ability to sleep [1]. According to research results, 98% of pregnant women reported poor sleep quality. Likewise, studies highlighted a relationship between changes in sleep quality and PSQI score.

In their study on sleep patterns and problems in pregnant women, Mindell et al. (2015) found that 38% of 2427 participants reported inadequate sleep [19]. In the same study, it was found that sleep problems increased as the trimester of pregnancy progressed. The prevalence of insomnia in pregnant women in European countries varies between 52% and 61% [20,21]. The higher rate of pregnant women in our study compared to other studies is due to the fact that they were in advanced gestational weeks and had a pregnancy complication such as gestational cholestasis.

A negative correlation was observed in our study between pregnant women’s weight gain and PSQI scores.

In their study analyzing sleep quality in pregnant women in China, Lyu et al. (2020) found that poor sleep quality in the middle or late pregnancy period was related to excessive gestational weight gain [22].

It has been reported in the literature that sleep problems of high-risk pregnant women are associated with the physical symptoms experienced and the gestational week [23]. Fluctuations in hormone levels, physical discomfort, and other pregnancy-related stress factors are considered to lead to excessive or inadequate sleep [24]. Since general sleep problems are a major public health issue worldwide, they should be identified given their adverse effects on risky pregnancies for both mothers and infants. In order to provide effective methods to improve sleep quality and maternal–fetus health, it is important to investigate sleep disorders caused by pregnancy complications within the pregnancy care.

Our findings demonstrated that there was a relationship between the hospitalization status of pregnant women and their PSQI scores. Pregnancy complications bring along feelings of fear and frustration, especially during hospitalization. Prenatal anxiety and depression symptoms must be dealt with with care, considering that they are frequently encountered in obstetric inpatients and have the potential to raise the risk of postpartum depression, thereby adversely influencing infant and child development [25]. Risky pregnancy can cause women to experience their pregnancy negatively and want the pregnancy process to end as soon as possible [16]. This situation explains the negative perception of pregnancy and sleep problems due to risky pregnancy. In this context, the literature and our study results are consistent.

Our findings demonstrated that there was an association between previous modes of delivery and changes in body image and the Physical Health sub-dimension, having health insurance and the Psychological Health sub-dimension, and age and the Environmental Health sub-dimension. Furthermore, considering pregnancy again was found to be associated with Physical Health, Psychological Health, and Social Relationships sub-dimensions and hospitalization status with Physical Health, Social Relationships, and Environmental Health sub-dimensions. Dağlar et al. (2019) found that the quality of life of third-trimester pregnant women was affected by their perception of health status, education level, number of pregnancies and births, perception of economic status, and readiness for parenting role [26]. Women who give birth vaginally have a higher quality of life than those who give birth by cesarean section [27]. In Aydın’s study, pregnant women with cholestasis stated that they experienced stress about the future pregnancy plan due to the problems they experienced and that they did not think about pregnancy for now [28]. Prior studies reported that women who experience high-risk pregnancies choose more reliable contraceptive methods afterwards [29].

In a qualitative study on pregnant women with cholestasis, women reported that itching increased at night and did not ease no matter what they did, and they could hardly sleep [28]. Saadati et al. (2018) found that 94% of high-risk pregnant women had sleep problems and this had a close negative relationship with quality of life [30].

The literature and our findings match in this way. It is obvious that itching, which is the most common dermatologic complaint of pregnant women with cholestasis, followed by fatigue, depressive symptoms, and sleep disorders, negatively affects women’s lives and worsens their quality of life. Even in uncomplicated pregnancies, quality of life decreases as the pregnancy progresses. The reason why the hospitalization status was found to be related to three sub-dimensions of quality of life (Physical Health, Social Relationships, and Environmental Health) is thought to be due to separation from family, spouse-child, and habitual environment along with uncertainty about pregnancy, and concerns about the health status of the fetus or herself.

### Limitations

Since cholestasis of pregnancy is a rare health problem, the researchers had difficulty in reaching women with cholestasis of pregnancy in the health institution. The findings of the study cannot be generalized due to the limited sample group in which volunteers participated in a single training and research hospital. Therefore, the findings can only be evaluated regarding the study participants, but they can be transferred to similar settings. It is recommended that the study be conducted in more than one location and center with a larger participant group.

## 5. Conclusions

Cholestasis of pregnancy creates problems such as pruritus, body image changes, hospitalization, and low sleep quality in women. Findings of this study showed that sleep quality as well as quality of life were low in pregnant women with cholestasis. Itching and related factors are not sufficiently considered and questioned in pregnancy care, especially in pregnant women with cholestasis. This study shows that sleep quality, prenatal comfort levels, and quality of life of pregnant women with cholestasis are affected. In addition, it is seen that women with this problem do not want to become pregnant again.

Midwives have an important role in helping women with cholestasis of pregnancy to cope with their experiences. They should be aware of other problems, especially itching, brought on by cholestasis, and provide support to the woman accordingly. Pregnant women need to be educated and supported in terms of nutrition, exercise, having the same sleeping hours, and controlling stress to enhance their quality of life.

This study underscores the necessity for healthcare providers, particularly midwives, to adopt a holistic care approach that addresses both the physical and psychological impacts of cholestasis of pregnancy. By incorporating strategies that emphasize patient education on nutrition, exercise, and stress management, healthcare professionals can significantly improve the pregnancy experience for women suffering from this condition. Further research in diverse and larger populations is crucial to enhance our understanding of cholestasis of pregnancy and to refine care practices, ultimately supporting the well-being of affected women more effectively.

## Figures and Tables

**Table 1 healthcare-12-01399-t001:** Pregnant Women’s Descriptive Characteristics.

Variables		n	%
Education Status	Literate	44	29.3
	Primary education	67	44.7
	High school and above	39	26.0
Employment Status	Yes	30	20.0
	No	120	80.0
Health Insurance Status	Yes	89	59.3
	No	61	40.7
Family type	Nuclear family	121	80.7
	Extended family	29	19.3
Number of Pregnancy	Primiparous	28	18.7
	Multiparous	122	81.3
Previous Mode of Birth	Vaginal	105	84.7
	Cesarean	19	15.3
Thinking about pregnancy again	Yes	33	22.0
	No	117	78.0
Hospitalization status	Yes	28	18.7
	No	122	81.3
Body Image Change	Yes	144	96.0
	No	6	4.0
Poor Sleep Quality	Yes	147	98.0
	No	3	2.0
Impact on social life	Yes	150	100.0
	No	0	0.0
Taking a warm shower	Yes	149	99.3
	No	1	0.7
Ice pack application	Yes	48	32.0
	No	102	68.0
Cream	Yes	125	83.3
	No	25	16.7
Cologne	Yes	58	38.7
	No	92	61.3
Yogurt	Yes	46	30.7
	No	104	69.3
Trying not to think	Yes	71	47.3
	No	79	52.7
Age, mean ± Standard deviation Median (Min–Max)	27.79 ± 6.33 27.00 (16.00–45.00)		
Obstetric history			
Number of pregnancies mean ± Standard deviation	3.71 ± 2.14 3.50 (1.00–10.00)		
Median (Min–Max)	2.39 ± 1.87 2.00 (0.00–9.00)		
Mean number of births ± Standard deviation	0.33 ± 0.70 0.00 (0.00–4.00)		
Median (Min–Max)	31.45 ± 3.41 32.00 (21.00–38.00)		

Min: Minimum, Max: Maximum.

**Table 2 healthcare-12-01399-t002:** Mean Scores the Pregnant Women Obtained From The Overall PCS, PSQI, and WHOQOL-BREF and Their Subscales.

Scales	Mean ± SD	Median (Min–Max)
PCS Total Score	61.20 ± 5.84	62.50 (44.00–72.00)
PCS subscales		
Husband	15.94 ± 2.30	16.00 (9.00–20.00)
Fetus	8.49 ± 1.32	9.00 (3.00–10.00)
People	12.12 ± 1.66	12.00 (7.00–15.00)
Mother	12.35 ± 1.59	13.00 (5.00–15.00)
Myself	12.30 ± 1.52	12.00 (7.00–15.00)
PSQI Total Score	9.52 ± 3.02	10.00 (4.00–17.00)
Physical Health	10.63 ± 2.18	10.86 (5.14–15.43)
Psychological Health	10.48 ± 2.10	10.67 (4.00–16.00)
Social Relations	11.31 ± 3.28	12.00 (4.00–18.67)
Environmental Health	11.27 ± 2.10	11.56 (5.33–16.44)

Mean: Average, SD: Standard Deviation, Min: Minimum, Max: Maximum; PCS: Prenatal Comfort Scale; PSQI: Pittsburgh Sleep Quality Index; WHOQOL-BREF: World Health Organization Quality of Life-Brief Form.

**Table 3 healthcare-12-01399-t003:** Relationship between Scale Scores and Some Variables.

Scales		Age	Cholestasis Diagnosis Week	Weight Gained
PCS Total	r	−0.142	0.019	−0.073
	*p*	0.084	0.818	0.378
PCS subscales				
Husband	r	−0.075	0.067	−0.014
	*p*	0.365	0.412	0.862
Fetus	r	−0.055	0.062	0.098
	*p*	0.503	0.451	0.238
People	r	−0.056	0.033	−0.039
	*p*	0.500	0.691	0.636
Mother	r	−0.180	0.022	−0.107
	*p*	0.027	0.785	0.197
Myself	r	−0.166	−0.079	−0.229
	*p*	0.042	0.339	0.005
PSQI	r	0.088	−0.051	−0.243
	*p*	0.287	0.532	0.003
Physical Health	r	0.016	0.130	−0.104
	*p*	0.842	0.112	0.207
Psychological Health	r	0.124	0.147	−0.074
	*p*	0.132	0.073	0.369
Social Relations	r	0.085	0.030	−0.127
	*p*	0.299	0.713	0.123
Environmental Health	r	0.194	0.142	−0.025
	*p*	0.017	0.083	0.762

r: Correlation coefficient.

**Table 4 healthcare-12-01399-t004:** Comparisons of Some Variables with PSQI.

Variables		PSQI		*p*
		Mean ± SD	Median (Min–Max)	
Education Status	Literate	8.77 ± 3.09	8.00 (4.00–15.00)	0.133 ^a^
	Primary education	9.94 ± 2.71	10.00 (5.00–15.00)	
	High School and Above	9.64 ± 3.34	9.00 (4.00–17.00)	
Employment Status	Yes	10.03 ± 3.41	10.00 (4.00–17.00)	0.341 ^b^
	No	9.39 ± 2.91	10.00 (4.00–16.00)	
Health Insurance Status	Yes	9.55 ± 2.92	9.00 (4.00–17.00)	0.905 ^b^
	No	9.48 ± 3.18	10.00 (4.00–16.00)	
Family Type	Nuclear Family	9.70 ± 3.06	10.00 (4.00–17.00)	0.151 ^b^
	Extended Family	8.76 ± 2.73	9.00 (4.00–14.00)	
Number of Pregnancy	Primiparous	10.11 ± 2.78	10.00 (4.00–15.00)	0.234 ^b^
	Multiparous	9.39 ± 3.07	9.50 (4.00–17.00)	
Previous Mode of Birth	Vaginal Birth	9.26 ± 3.13	9.00 (4.00–17.00)	0.233 ^b^
	Cesarean section	10.11 ± 2.85	10.00 (5.00–14.00)	
Considering Pregnancy Again	Yes	8.67 ± 2.78	8.00 (4.00–15.00)	0.066 ^b^
	No	9.76 ± 3.05	10.00 (4.00–17.00)	
Hospitalization	Yes	8.36 ± 2.93	8.50 (4.00–14.00)	0.025 ^b^
	No	9.79 ± 2.99	10.00 (4.00–17.00)	
Body Image Change	Yes	9.53 ± 3.06	10.00 (4.00–17.00)	0.769 ^b^
	No	9.17 ± 2.04	9.00 (7.00–12.00)	
Change in Sleep Quality	Yes	9.45 ± 3.00	10.00 (4.00–17.00)	0.035 ^b^
	No	13.00 ± 1.00	13.00 (12.00–14.00)	

Mean, SD: Standard Deviation, ^a^: Kruskal–Wallis H test, ^b^: Mann–Whitney U test; PSQI: Pittsburgh Sleep Quality Index.

**Table 5 healthcare-12-01399-t005:** Comparisons of Some Variables with Quality of Life Scale.

Variables		Physical Health	Psychological Health	Social Relations	Environmental Health
		Mean ± SD			
Education Status	Literate	10.38 ± 2.14	10.12 ± 2.02	10.58 ± 3.67	10.83 ± 2.36
	Primary education	10.77 ± 2.30	10.73 ± 2.20	11.84 ± 3.42	11.57 ± 1.93
	High School and Above	10.68 ± 2.05	10.44 ± 2.00	11.25 ± 3.04	11.26 ± 2.04
	*p*-value	0.591 ^a^	0.369 ^a^	0.092 ^a^	0.428 ^a^
Employment Status	Yes	10.63 ± 1.88	10.73 ± 1.95	11.78 ± 3.34	11.44 ± 2.23
	No.	10.63 ± 2.26	10.41 ± 2.14	11.20 ± 3.26	11.23 ± 2.07
	*p*-value	0.899 ^b^	0.893 ^b^	0.375 ^b^	0.636 ^b^
Health Insurance Status	Yes.	10.52 ± 2.26	10.07 ± 2.20	11.06 ± 3.24	11.23 ± 2.05
	No	10.78 ± 2.07	11.07 ± 1.81	11.70 ± 3.31	11.34 ± 2.18
	*p*-value	0.503 ^b^	0.007 ^b^	0.235 ^b^	0.842 ^b^
Family Type	Nuclear Family	10.67 ± 2.18	10.47 ± 2.16	11.40 ± 3.24	11.36 ± 2.12
	Extended Family	10.44 ± 2.20	10.48 ± 1.88	10.94 ± 3.44	10.90 ± 1.98
	*p*-value	0.660 ^b^	0.909 ^b^	0.481 ^b^	0.300 ^b^
Number of Pregnancy	Primiparous	10.71 ± 2.48	9.95 ± 2.27	11.14 ± 3.04	11.06 ± 2.21
	Multiparous	10.60 ± 2.12	10.60 ± 2.05	11.36 ± 3.34	11.32 ± 2.08
	*p*-value	0.586 ^b^	0.270 ^b^	0.710 ^b^	0.681 ^b^
Previous Mode of Birth	Vaginal	10.74 ± 2.17	10.60 ± 2.05	11.37 ± 3.40	11.25 ± 2.05
	Cesarean	9.74 ± 1.52	10.35 ± 1.99	10.74 ± 3.13	11.60 ± 2.27
	*p*-value	0.043 ^b^	0.292 ^b^	0.410 ^b^	0.416 ^b^
Considering Pregnancy Again	Yes	9.77 ± 2.18	9.74 ± 2.43	9.74 ± 3.39	10.76 ± 2.49
	No	10.87 ± 2.13	10.68 ± 1.96	11.67 ± 3.12	11.42 ± 1.96
	*p*-value	0.010 ^b^	0.031 ^b^	0.002 ^b^	0.222 ^b^
Hospitalization	Yes	9.78 ± 2.50	10.19 ± 2.41	9.90 ± 3.51	10.51 ± 2.33
	No	10.82 ± 2.06	10.54 ± 2.03	11.64 ± 3.15	11.45 ± 2.00
	*p*-value	0.036 ^b^	0.554 ^b^	0.019 ^b^	0.042 ^b^
Body Image Change	Yes	10.72 ± 2.13	10.50 ± 2.11	11.29 ± 3.25	11.26 ± 2.11
	No	8.48 ± 2.38	10.00 ± 1.89	12.00 ± 4.04	11.48 ± 1.89
	*p*-value	0.028 ^b^	0.626 ^b^	0.493 ^b^	0.784 ^b^
Change in Sleep Quality	Yes	10.61 ± 2.18	10.45 ± 2.07	11.26 ± 3.26	11.21 ± 2.05
	No	11.62 ± 2.58	11.56 ± 3.91	14.22 ± 3.36	14.22 ± 2.78
	*p*-value	0.514 ^b^	0.893 ^b^	0.139 ^b^	0.072 ^b^

Mean: Average, SD: Standard Deviation, ^a^: Kruskal–Wallis H test, ^b^: Mann–Whitney U test.

## Data Availability

The data presented in this study are available on request from the corresponding author.

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
