# Peer review of "Evaluation of Prenatal Comfort, Sleep, and Quality of Life in Pregnant Women with Cholestasis: A Cross-Sectional Study"

_healthcare, 2024, doi:10.3390/healthcare12141399_

Round 1

Reviewer 1 Report

Comments and Suggestions for Authors

Review of the manuscript: 

Thank you for the opportunity to review the manuscript Evaluation of Prenatal Comfort, Sleep and Quality of Life in Pregnant Women with Cholestasis: A Cross-Sectional Study. The manuscript covers an important topic for public health. Namely, the study evaluates prenatal comfort, sleep, and quality of life in pregnant women with cholestasis

After thoroughly evaluating the manuscript, I have several comments and suggestions that I believe will enhance its clarity, rigour, and impact.

Introduction

The introduction needs to be completely reorganized. I recommend first introducing readers to cholestasis, its prevalence, pathophysiology, the clinical picture, and possibly therapy. Following that, discuss the consequences or complaints arising from cholestasis, such as discomfort (describing it in detail), sleep disorders (specifying the type), and impaired quality of life (identifying the affected domains), all referenced to previous research.

It is also suggested that there is no need to define the quality of life, as it is a well-researched concept. In lines 40-41, where it states “in our country,” it is necessary to specify which country is being referred to, as the reader may not be aware.

Materials and Methods

Population and Sample

Why were 150 pregnant women selected? How was the required sample size calculated? State whether the sample is homogeneous in terms of duration of cholestasis. Also, it is stated that the investigation was conducted in an outpatient clinic, and later, it mentions hospitalization. Were women hospitalized during cholestasis therapy? Can therapy affect comfort, quality of sleep and life in general?

Data Collection

All instruments are well described and tested in the Turkish cultural environment. It is necessary to add a description of the sociodemographic and obstetric variables defined in the Participant Information Form.

Results

The results are presented in textual and tabular form, but they need to be redesigned because they are rather cluttered and difficult to follow and read.

The descriptive presentation of pregnant women should be separated into two tables, one for sociodemographic variables and the other for obstetric variables. If the authors still want one table, I recommend a clear demarcation of these two groups of variables.

Table 2 requires a simpler presentation. It is not necessary to insert lines explaining that something is a subscale (this applies in particular to WHOQoL). The table should be designed so that there is a clear demarcation between the three observed instruments. For tables 3-5, specify the title. What are quantitative and qualitative variables? They are not mentioned anywhere in the methodology, only sociodemographic and obstetric variables. In Table 5, delete the column with Mean±SD and make a row under the variables (as in Table 4).

Discussion

In the discussion, the authors analyzed the results obtained following the existing literature, which is insufficient regarding sleep quality, life, and comfort in pregnant women with cholestasis. Potential limitations are noted, and conclusions are clear.

Most references are up-to-date and relevant to the topic. Still, it is necessary to standardize the journal titles (obey the instructions for authors), not to state the full journal name, and abbreviate it somewhere.

Author Response

Dear Reviewer,

Thank you very much for your precious comments and contributions.

We really appreciate it.  Please see the attachment.

Best regards.

Reviewer 2 Report

Comments and Suggestions for Authors

As attached

Comments on the Quality of English Language

Minor- moderate corrections

Author Response

Dear reviewer,

Dear Reviewer,

Thank you very much for your precious comments and contributions.

We really appreciate it. Please see the attachment.

Best regards.

Reviewer 3 Report

Comments and Suggestions for Authors

Thanks to the authors

What sampling method was used?

Why is there no connection between quality of life, quality of sleep and comfort?

It is better to write the discussion based on the order of the findings

Author Response

(The authors gave the same response as above.)

Round 2

Reviewer 1 Report

Comments and Suggestions for Authors

The authors accept all the suggestions given, so the revised version of the manuscript titled Evaluation of Prenatal Comfort, Sleep and Quality of Life in Pregnant Women with Cholestasis: A Cross-Sectional Study is now easy to read, and all parts of the manuscript are better presented, which has improved the overall quality of the manuscript.

Considering the improvements and the significance of the topic under examination, I recommend that the manuscript be accepted.